# The Extracellular Mass to Body Cell Mass Ratio as a Predictor of Mortality Risk in Hemodialysis Patients

**DOI:** 10.3390/nu14081659

**Published:** 2022-04-15

**Authors:** Mar Ruperto, Guillermina Barril

**Affiliations:** 1Department of Pharmaceutical & Health Sciences, Faculty of Pharmacy, Universidad San Pablo-CEU, CEU Universities, Urbanización Montepríncipe, Alcorcón, 28925 Madrid, Spain; 2Grupo USP-CEU de Excelencia “Nutrición para la vida (Nutrition for Life)”, Ref: E02/0720, Alcorcón, 28925 Madrid, Spain; 3Nephrology Department, Hospital Universitario La Princesa, 28006 Madrid, Spain; gbarril43@gmail.com

**Keywords:** bioelectric impedance analysis, extracellular mass to body cell mass ratio, fluid overload, interleukine-6, haemodialysis, protein-energy wasting

## Abstract

The extracellular mass/body cell mass ratio (ECM/BCM ratio) is a novel indicator of nutritional and hydration status in hemodialysis (HD) patients. This study aimed to explore the ECM/BCM ratio as a predictor of mortality risk with nutritional-inflammatory markers in HD patients. A prospective observational study was conducted in 90 HD patients (male: 52.2%; DM: 25.60%). Clinical and biochemical parameters [serum albumin, serum C-reactive protein (s-CRP), interleukine-6 (IL-6)] were analysed and bioelectrical impedance analysis (BIA) was performed. Protein-energy wasting syndrome (PEW) was diagnosed using malnutrition-inflammation score (MIS). Based on BIA-derived measurements, the ECM/BCM ratio with a cut-off point of 1.20 was used as a PEW-fluid overload indicator. Comorbidity by Charlson index and hospital admissions were measured. Out of 90 HD patients followed up for 36 months, 20 patients (22.22%) died. PEW was observed in 24 survivors (34.28%) and all non-survivors. The ECM/BCM ratio was directly correlated with MIS, s-CRP, Charlson index and hospital admissions but was negatively correlated with phase angle and s-albumin *(all*, *p* < 0.001). Values of the ECM/BCM ratio ≥ 1.20 were associated with higher probability of all-cause mortality (*p* = 0.002). The ECM/BCM ratio ≥ 1.20, IL-6 ≥ 3.1 pg/mL, s-CRP and s-albumin ≥ 3.8 g/dL and Charlson index were significantly associated with all-cause mortality risk in multivariate adjusted analysis. This study demonstrates that the ECM/BCM ratio ≥ 1.20 as a nutritional marker and/or fluid overload indicator had a significant prognostic value of death risk in HD patients.

## 1. Introduction

Protein-energy wasting syndrome (PEW), inflammation and fluid overload are conditions that either alone or together are associated with increased morbidity and mortality in hemodialysis (HD) patients [1,2,3]. The diagnosis of PEW [4] combines certain nutritional markers in three of the four proposed categories (biochemical data, body mass, muscle mass and dietary intake), while the diagnosis of inflammation and volume overload status also requires inflammatory biomarkers and other techniques such as bioelectrical impedance analysis (BIA). Potential causative PEW factors such as uremic toxicity, protein and amino acid losses during dialysis, metabolic acidosis, insulin resistance as well as inadequate energy and protein intake are related to the underlying inflammatory state, the latter contributing to 30–50% of PEW on dialysis [2,5].

Fluid overload is a common feature among dialysis patients, with an estimated incidence of 56.5% to 73.1% [6,7]. Epidemiological studies [7,8,9] have shown that fluid overload is a recognized independent predictor of adverse clinical outcomes and all-cause death risk. An impaired volume status is associated with high blood pressure (hypertensive status), left ventricular hypertrophy and cardiac dysfunction causes, oedemas and intradialytic hypotension secondary to the high rate of ultrafiltration to remove excess fluid during the dialysis session [7,8,9,10].

Fluid optimization is one of the main goals in HD treatment as it relates to dialysis adequacy, intra-dialysis symptoms and post-dialysis dry weight. Traditionally, adjustment of fluid status is done by clinical examination at the discretion of the practitioner (jugular ingurgitation, peripheral oedemas, dyspnea, etc.), monitoring of blood pressure (BP) and target dry weight. As it stands, there is no gold standard for assessing the volume status and changes in the distribution of the extracellular (ECW) and intracellular (ICW) body compartments, as well as their complex relationship with albuminemia and inflammatory status; therefore, more objective techniques and/or markers are needed to diagnose the triad of PEW, inflammation, and fluid overload among HD patients.

The bioelectric impedance analysis (BIA) is a simple, easy-to-use and reproducible measurement tool that has been validated for the assessment of hydration status and body composition in dialysis [11,12]. Two meta-analyses [13,14] showed that BIA-derived measures of hydration (e.g., ICW, ECW, total body water (TBW) and their ratios) were independent predictors of cardiovascular (CV) events and mortality risk in patients receiving dialysis.

The ratio of extracellular cell mass to body cell mass (ECM/BCM ratio) has emerged as a sensitive indicator of nutritional and hydration status in dialysis patients [15,16,17] as well as in the population not undergoing dialysis [18,19,20,21]. ECM is related to body fluids (e.g., plasma volume, extracellular water) and is considered a non-metabolically active body component. Conversely, BCM constitutes the metabolically active body mass responsible for energy exchange (e.g., muscle mass). An ECM/BCM ratio cut-off point ≥ 1.2 was suggested as a discriminator of nutritional and/or hydration status in HD patients [15,16,17]. High values of the ECM/BCM ratio were associated with PEW, systemic inflammation, volume overload, CV events and adverse prognosis in dialysis [15,16,17]. To date, only a few studies have focused on the prognostic value of the ECM/BCM ratio as a predictor of mortality in HD patients. This study aimed to explore the prognostic value of the ECM/BCM ratio as a predictor of mortality risk with nutritional-inflammatory markers in HD patients.

## 2. Patients and Methods

### 2.1. Patient Population

A prospective observational study was carried out at the Hospital Universitario La Princesa (Madrid, Spain) from March 2016 to January 2020. A sample of 190 HD patients were enrolled from one single dialysis unit (Figure 1). Patients aged over 65 years were eligible if they were on HD treatment and were found to be stable in the last three months prior to inclusion. Exclusion criteria were advanced heart failure, chronic pulmonary disease, cirrhosis, and certain comorbidities (infection or malign tumour); medications (NSAIDs, corticosteroid or diuretics); surgeries interfering with nutritional status; nutritional support (oral nutritional supplements, enteral or parenteral nutrition); patients with leg amputations, pacemakers or joint prostheses; and life expectancy less than 3 months. At baseline, 29 HD patients did not meet the inclusion criteria for participation in this study. Participants were censored on transplantation, change of dialysis modality and transfer to another centre. The final sample size was 90 HD patients (Figure 1). This study was conducted in accordance with the Helsinki Declaration and Good Clinical Practice Guidelines. Written informed consent was obtained from all participants before onset of this study. Ethical approval and permission to conduct this study were obtained from the local Ethic Committee (code number. 681).

### 2.2. Data Collection

Clinical, demographic, nutritional and biochemical data were evaluated. Data on aetiology of CKD, vascular access, major comorbidities and time on HD (months) were collected. Dialysis adequacy was calculated by the dialysis dose administered (Kt/V *urea*) using a single pool (*sp*) urea kinetic model (second generation Kt/V Daurgidas) with the blood urea concentration before and after dialysis in the mid-week HD session, as well as the urea reduction rate (URR), expressed as a percentage.

Participants were undergoing at least 4 h dialysis sessions three times per week by using high-flux dialysis membranes. Blood pressure (BP) was measured in the supine position post-HD treatment. The Charlson comorbidity index (Charlson index) questionnaire without age and CKD components was used [22]. The number of hospital admissions during the 36 months of follow-up was registered and the mean number of admissions per year was calculated. The cause of death was categorized based on the International Classification of Diseases 10th edition (ICD-10) in the following areas: infectious diseases (including sepsis), cardiovascular diseases, gastrointestinal diseases (including gastrointestinal bleeding), neurologic diseases (including stroke), musculoskeletal diseases and other causes. Mortality data were also double-checked retrospectively from medical records.

### 2.3. Anthropometric Measures and Body Composition Analysis

Anthropometric measurements were obtained after HD treatment (15–30 min after the HD session) at dry weight. Height, post-dialysis dry weight and body mass index (BMI) were recorded. Body composition was analysed with a portable bioelectrical impedance analyser (BIA-101^®®^. Akern-RJL Systems, Florence, Italy), at a frequency of 50 kHz and 800 μA constant electrical flow through the body in horizontal supine position (30 min after the mid-week HD session). Tetrapolar distal validated method [23] was used by placing two pairs of disposable electrodes (BiatrodesTM 100’S, Akern, Florence, Italy) on the right hand and foot, and whole-body BIA was performed on the side contralateral to the vascular access. Exchange Na/K, total body water (TBW), ECW, ICW, fat-free mass (FFM), muscle mass (MM), BCM and phase angle (PA) were measured and analysed by BodygramPro v.3.0 software based on the manufacturer’s standards. Results were compared with the 50th percentile population values as the standard reference for the assessment of hydration status and body composition measurements, as was previously used as a reference method for dialysis patients [12].

Extracellular cell mass (ECM) was calculated as follows: ECM = FFM-BCM, where FFM is fat-free mass, and BCM, body cell mass, both in kg. The ECM/BCM ratio was calculated by dividing ECM by BCM, both expressed in kg. According to previous studies, a cut-off point of the ECM/BCM ratio ≥ 1.20 was an independent predictor of PEW and fluid overload in HD patients [16,17] and in peritoneal dialysis patients [15]. The ECM/BCM ratio ≥ 1.20 was considered a PEW-fluid overload marker in this study.

### 2.4. Laboratory Parameters

Blood samples were collected at the time of monthly sampling and immediately prior to the onset of mid-week dialysis sessions. S-albumin was analysed by the colorimetric standard method (Roche/Hitachi 904^®®^/Modular P: ACN 0413; Roche Diagnostics, Basel, Switzerland.) using the bromocresol green method [24]. Based on the diagnostic criteria for PEW [4], the cut-off point for s-albumin was set at 3.8 g/dL. Serum transferrin and serum C-reactive protein (s-CRP) were analysed by immunoturbidimetry methods (Roche/Hitachi 904^®®^/Model P:ACN 218, Roche Diagnostics, Basel, Switzerland). The concentration of s-CRP (non-high sensitivity) was considered an inflammatory marker set at a cut-off point at or above 1 mg/dL. Plasma IL-6 was measured by high sensitivity ELISA (Human IL-6 Quantikine HS ELISA kit; R&D Systems, Minneapolis, MN, USA) using a four-plate ELISA DSXTM processor (Dynex, Chantilly, VA, USA). Intra-assay IL-6 precision was 0.7 pg/mL. Median values of plasma IL-6 in the sample were used to define the cut-off point set at 3.1 pg/mL.

### 2.5. Malnutrition-Inflammation Score Questionnaire

Nutritional status was assessed by the malnutrition-inflammation score (MIS) questionnaire [25], which included 10 components: 7 subjective (concerning the patient’s medical history and physical examination) and 3 objective parameters (s-albumin, total binding iron capacity and body mass index (BMI). The MIS components range from 0 (normal) to 30 (severely wasted). In agreement with other studies, PEW was defined as a value of MIS ≥ 5 points [26,27].

### 2.6. Statistical Analysis

Continuous variables were expressed as mean and standard deviation, with the comparisons between groups performed by the unpaired Student’s *t*-test for normally distributed variables and Fisher’s exact test for non-normally distributed variables. Categorical variables were described using proportions and were analysed by the chi-square test. Correlations were tested by bivariate Pearson correlations for continuous variables. The Kaplan–Meier method [28] was used to calculate cumulative survival probabilities according to the cut-off point of ECM/BCM ratio ≥ 1.20, and the difference between survival curves was assessed by the log-rank test. Univariate and multivariate Cox proportional hazard regression analyses were used to evaluate the prognostic value of the ECM/BCM ratio and the risk of death. The variables that significantly affected survival in the univariate analysis were subsequently tested in multivariate models using a forward stepwise procedure with a probability to entry of 0.05 and a probability to removal of 0.20. The multivariate Cox proportional hazard model was adjusted for age, gender and time on HD as potential confounders. The adjusted hazard ratio (aHR) at 95% confidence interval (95%CI) was calculated. Statistical analyses were performed by using SPSS version 24.0 (IBM Corp., Armonk, NY, USA) software. *p*-values < 0.05 were set as significant.

## 3. Results

### 3.1. Global Data and Comparison between Survivor and Non-Survivors

Out of 90 HD patients, 20 patients (22.22%) died during the 36-month follow-up. Diabetes mellitus (25.6%) was the main aetiology of chronic kidney disease (CKD). The causes of death were cardiovascular disease (*n* = 4), sudden cardiac death (*n* = 3), cerebrovascular disease (*n* = 2), infection (*n* = 4), neoplasm (*n* = 5) and unknown cause (*n* = 2). Demographic and clinical characteristics of the 90 HD patients are summarized in Table 1. Gender, age, diabetes mellitus (DM), adequacy of dialysis [Kt/V urea (*sp*), URR] did not differ significantly between survivors and non-survivors. PEW was found in 24 survivors (34.28%), while all non-survivors were wasted (MIS values ≥ 5 points).

Table 2, shows body composition and laboratory data of survivor and non-survivor groups. Survivors had a significantly lower exchange Na/K, TBW and s-CRP than no-survivor group (*at least*, *p* < 0.05). BIA-derived body composition measurements, such as FFM and BCM, differed significantly between groups (*both*, *at least p* < 0.05).

Mean value of the ECM/BCM ratio was significantly higher in non-survivors than in survivors (1.46 ± 0.32 vs. 2.14 ± 0.23; *p* < 0.0001, respectively). The ECM/BCM ratio was positively correlated with MIS (r = 0.47; *p* < 0.001), s-CRP (r = 0.46; *p* < 0.001), Charlson index (r = 0.39; *p* < 0.001) and hospital admissions (r = 0.33; *p* = 0.001), but negatively correlated with PA (r = −0.72; *p* < 0.001) and s-albumin (r = −0.41; *p* < 0.001). As expected, the non-survivor group had significantly higher values of MIS and plasma IL-6, and lower s-albumin concentration compared with the non-survivor group (*all*, *at least*, *p* < 0.05) (Table 2).

### 3.2. Analysis of the ECM/BCM Ratio Cut-Off Point

The ECM/BCM ratio ≥ 1.20 was found in 45 survivors (50%) and the entire non-survivors’ group (*n* = 20). Analysing the ECM/BCM ratio cut-off point (<1.20 vs. >1.20) in the survivor group, significant differences with MIS (4.32 ± 3.31 vs. 10.05 ± 4.64), exchange Na/K (1.06 ± 0.17 vs. 1.47 ± 0.40), ECW (16.66 ± 2.56 vs. 20.01 ± 4.13), s-CRP (0.62 ± 0.37 vs. 1.05 ± 0.90) and hospital admissions (0.52 ± 0.82 vs. 1.28 ± 1.42) were found, respectively (*at least*, *p* < 0.01). Additionally, survivor HD patients with values of the ECM/BCM ratio ≥ 1.20 vs. < 1.20 points, had significantly lower PA (3.95 ± 0.73 vs. 5.32 ± 0.83) and s-albumin (3.65 ± 0.36 vs. 4.02 ± 0.45) (*all*, *p* < 0.001), while a nonsignificant trend of plasma IL-6 was found (*p* = 0.11) (data not shown).

### 3.3. Extracellular Mass to Body Cell Mass Ratio and Mortality

The predictive ability of the ECM/BCM ratio as an indicator of being wasted and fluid overload was previously described elsewhere by the authors [16]. Kaplan–Meier analysis showed that HD patients with the ECM/BCM ratio ≥ 1.20 had a higher probability of all-cause mortality (log-rank test x^2^ = 9.387; *p* = 0.002) as shown in Figure 2.

### 3.4. Cox proportional Hazards Analysis of Mortality

Univariate Cox regression analysis showed that all-cause mortality was positively associated with MIS, Na/K exchange, s-CRP levels, hospital admissions and Charlson index, while it was negatively associated with s-albumin in HD patients (Table 3).

Using multivariable Cox proportional hazard analysis after adjusting for potential confounders (gender, age and time on HD), it was shown that cut-off points of the ECM/BCM ratio ≥ 1.20, plasma IL-6 ≥ 3.1 pg/mL, s-CRP and Charlson index were significantly associated with all-cause mortality risk, while a significant negative association was found with s-albumin ≥ 3.8 g/dL (Table 4).

## 4. Discussion

Results from this study demonstrate that the ECM/BCM ratio as a nutritional marker and/or fluid overload indicator had significant prognostic value of mortality risk in a selected cohort of HD patients followed up to 36 months. One important finding of this study was that values of the ECM/BCM ratio ≥ 1.20 together with nutritional-inflammatory markers (s-albumin, s-CRP, IL-6) and comorbidities, independently predicted all-cause mortality after adjustment for potential confounders.

Protein-energy wasting, inflammation and fluid overload triad are well-known independent mortality predictors in dialysis patients [1,2,7,29]. In this study, gender, age, time on HD and DM did not differ significantly in survivor and non-survivor groups. Interestingly, PEW was also found in 34.3% of HD survivors and in all non-survivors (Table 1). High mean values of MIS and increased values of s-CRP and plasma IL-6 were found in the non-survivor group. Protein-energy wasting (PEW) syndrome is an often-comorbid condition diagnosed in up to one-third of dialysis patients [30]. Numerous studies [25,26,31] reported the use of MIS as a predictor of both mortality and hospitalization in HD patients. The MIS questionnaire was a significant predictor for mortality at 10 years [1] and the best predictive score for all-cause mortality and secondary endpoints such as CV events among HD patients [31]. In the current study, MIS (HR 1.125 95%CI, 1.036–1.221; *p* = 0.005) and hospital admissions (HR 1.405 95%CI, 1.131–1.746; *p* = 0.002) were significant predictors of mortality risk in univariate analysis, while no association was found in the adjusted Cox multivariate analysis. These results are consistent with other studies [25,26,31,32] showing that MIS is a significant and independent predictor of risk of death in dialysis.

Serum albumin is the most universal predictor of hospitalization and mortality in both CKD and dialysis patients [2,33,34,35,36]. According to the International Society of Renal Nutrition and Metabolism [4], a cut-off point of s-albumin < 3.8 g/dL is considered a diagnostic criterion for being wasted in CKD patients. In the present study, on comparing survivor and non-survivor groups, it was found that half of the non-survivors had s-albumin levels < 3.8 g/dL associated with the highest levels of s-CRP (1.33 ± 0.46), IL-6 (6.32 ± 5.33) and significantly greater number of hospital admissions (2.60 ± 2.36). In a prospective cohort study [35], s-albumin levels < 3.8 g/dL were a strong predictor of mortality associated with a moderate increase in interdialytic weight gained. Results obtained from this study, in adjusted analysis, showed that s-albumin ≥ 3.8 g/dL appears as an independent predictor of long-term survival. Therefore, our results also support the usefulness of s-albumin as a nutritional and/or a disease marker as well as a significant predictor of death and hospitalization risk.

Inflammation is an overlapping condition associated with higher morbidity and mortality risk in HD patients. Serum C-reactive protein (CRP) and interleukin-6 (IL-6) are known inflammation biomarkers and independent predictors of cardiovascular disease (CVD) and mortality risk in dialysis patients [5,37]. In this study, s-CRP and plasma IL-6 were shown to be significant independent predictors of all-cause mortality in both univariate and multivariate analyses. It was noted that plasma IL-6 levels > 3.1 pg/mL increased the aHR risk of death 2.853-fold, whereas s-CRP as an early acute phase reactant increased the death risk 1.109-fold. Some studies [2,37] have shown high morbidity and mortality rates in HD patients with increases in acute phase positive reactants (s-CRP, fibrinogen), proinflammatory cytokines (IL-1, IL-6, tumour necrosis factor alpha: TNF-α), and decreases in s-albumin, and transferrin level. A prospective longitudinal study [37] concluded that for each pg/L increase in IL-6 concentration, the HR for death from all-causes was 1.06 (95%CI 1.01 to 1.10) after adjustment for demographic and clinical parameters. Honda et al. [38] reported IL-6 to be a stronger predictor than s-CRP for all-cause and CVD mortality. A meta-analysis [5] showed that both s-CRP and IL-6 could significantly predict all-cause and CV disease mortality. In accordance with some studies [5,37,38], both IL-6 and s-CRP were significant predictive biomarkers for all-cause mortality to guide and monitor nutritional-inflammation status in dialysis.

Bioelectrical impedance analysis is a well-proven method for assessing both body composition and hydration status. Fluid overload is a common condition and mortality risk factor of adverse outcomes among dialysis patients. Dialysis adequacy and optimal distribution between body water compartments have attracted much interest because of their association with wasting, inflammation and dialysis survival outcomes. In the present study, BIA-derived body composition measurements (BCM, MM, PA) as well as hydration markers (exchange Na/K, ECW) were significantly different between survivor and non-survivor groups (Table 2). As expected, the non-survivor group had significantly lower values of BCM and PA compared to their survivor counterparts. Likewise, over 70% of non-survivors had PA values < 4°, whereas these values were present only in 24% of HD survivors (data not shown). Low BCM and PA have been identified as nutritional and prognosticator markers [39]. In a large HD cohort [40], a significant increase of death risk was reported among patients with PA < 4°, even after adjustment of several nutritional indicators. Phase angle and s-albumin, together with additional biochemical nutritional and inflammatory parameters (i.e., s-CRP, IL-6), may help to monitor and adjust dry body weight in wasted HD patients, since hypoalbuminemia is also caused by hemodilution during states of chronic volume expansion. In addition, in this study, mean BP measures and univariate regression showed, in particular DBP, to be a significant protective factor, in survivors and non-survivors. In fact, these results confirm that blood pressure control is a protective risk factor for mortality, in line with previous studies [41,42] in HD patients.

The ECM/BCM ratio has emerged as a novel sensitive index for diagnosing PEW and fluid overload, and a mortality predictor in dialysis patients [15,17]. In this study, the ECM/BCM ratio was positive and significantly correlated with nutritional-inflammatory parameters (MIS, s-CRP), comorbidities and hospital admissions (*all at least*, *p* < 0.05). The cut-off point of the ECM/BCM ratio (1.20) displayed significant differences with s-albumin, exchange Na/K and PA. The study’s main finding was that the ECM/BCM ratio ≥ 1.20 increased the adjusted HR death risk 5.078-fold (Table 4). Avram et al. [15] reported that the ECM/BCM ratio had a prognostic value for survival during an 8-year follow-up in peritoneal dialysis. In another study by our research group [16], we concluded that the ECM/BCM ratio ≥ 1.20 was a sensitive index discriminating nutritional status and/or fluid overload in HD patients, compared to age- and sex-matched controls.

This study has some strengths and limitations. First, it was a prospective cohort study conducted at a single HD centre. However, this study provided detailed information on nutritional and inflammatory status and body composition analysis measured by a validated BIA technique in dialysis patients [12]. Nevertheless, some of our results may be confounded by other factors that could not be fully taken into account in this study. Second, vascular access, especially the tunnelled central venous catheter, is a limiting factor in this study, since it is associated with dialysis adequacy and nutritional and inflammatory status as well as mortality in HD patients. Third, to date, BIA is currently one of the simpler methods used in the clinical practice, but only one single BIA measurement was assessed in this study. In addition, this study collected BIA-derived measurements along with nutritional-inflammatory markers such as s-albumin, s-CRP and plasma IL-6 that may be useful in achieving target dry weight, thereby improving dialysis outcomes. Fourth, the residual kidney function and dietary sodium intake were not registered in this study, which might be important modifiers of the associations studied herein. By contrast, most of HD patients included in this study were prevalent on HD treatment and had oligoanuria (<100 mL/urine volume/day) at the baseline. Fifth, cardiac imaging, functional testing or natriuretic peptides were not analysed, which could potentially be of interest for the exclusion of CV diseases. Conversely, BP measurements were used as a surrogate parameter within routine clinical examination and standard care in the HD unit. Sixth, our findings may not be wholly applicable to CKD or peritoneal dialysis patients, as volume status varies within these situations. Finally, to the best of our knowledge, there are no such published reports with regard to the prognostic value of the ECM/BCM ratio, in a prospective cohort study in HD patients.

## 5. Conclusions

This study demonstrates that the ECM/BCM ratio as a nutritional marker and/or fluid overload indicator had a significant prognostic value of mortality risk in HD patients. The results from this study revealed that values of the ECM/BCM ratio ≥ 1.20 together with nutritional-inflammatory markers (s-albumin, s-CRP, IL-6) and comorbidities independently predicted all-cause mortality after adjustment for potential confounders. Considering those findings, we emphasise the need for further studies assessing the ECM/BCM ratio as a nutritional and/or fluid indicator in long-term studies in HD patients.

## Figures and Tables

**Figure 1 nutrients-14-01659-f001:**
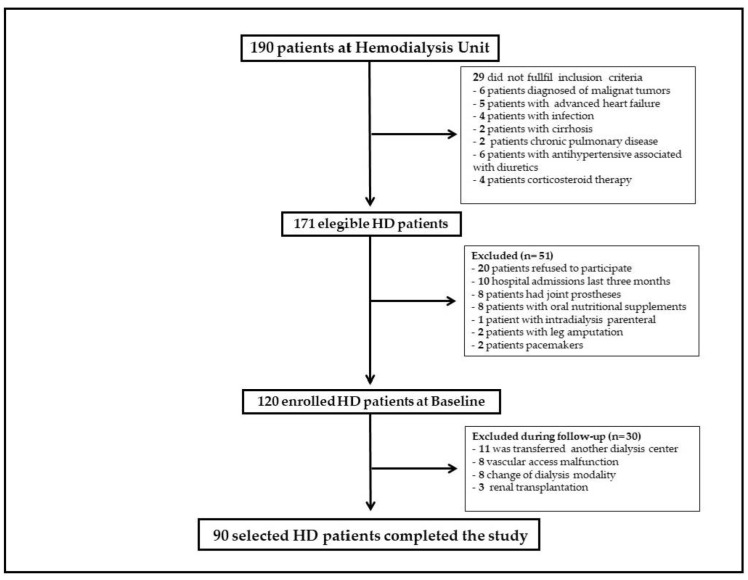
Flow chart of the recruitment and selection of hemodialysis patients in the study.

**Figure 2 nutrients-14-01659-f002:**
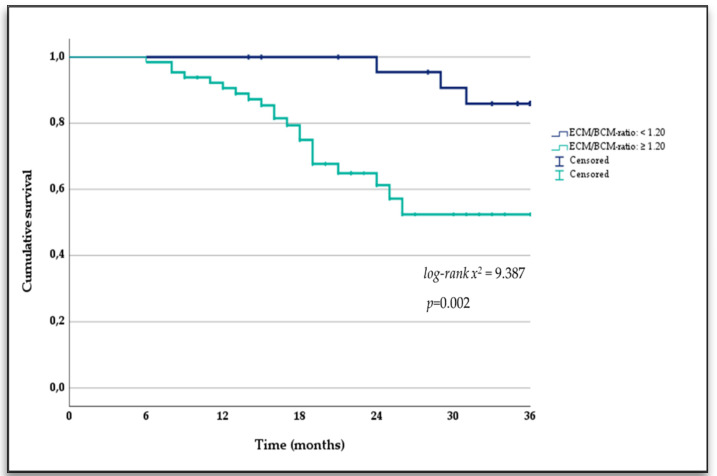
Kaplan–Meier survival curves of the study participants stratified by the cut-off-point of the extracellular mass to body cell mass ratio as a mortality risk predictor in hemodialysis patients.

**Table 1 nutrients-14-01659-t001:** Demographic and clinical characteristics of 90 hemodialysis patients.

Variable	Total (*n* = 90)	Survivors (*n* = 70)	Non-Survivors (*n* = 20)	*p*-Value
Gender (Male) *n* (%)	47 (52.20)	35 (50.0)	12 (60.0)	0.43
Age (years)	75.91 ± 3.97	75.77 ± 3.89	76.40 ± 4.28	0.53
Time on HD (months)	36.57 ± 38.70	37.19 ± 36.70	34.40 ± 41.20	0.77
DM *n*; (%)	23 (25.60)	14 (20.0)	9 (45.0)	0.20
SBP (mmHg)	125.74 ± 20.02	128.11 ± 19.07	118.40 ± 21.61	0.059
DBP (mmHg)	71.46 ± 12.47	73.55 ± 11.68	65.00 ± 12.92	0.007
Catheter *n* (%)	48 (55.80)	30 (45.50)	18 (90.0)	<0.001
Kt/V *urea* (sp)	1.33 ± 0.44	1.32 ± 0.46	1.43 ± 0.31	0.50
URR (%)	67.12 ± 12.05	66.04 ± 12.40	72.17 ± 9.07	0.11
Hospital admissions (number/per year)	1.07 ± 1.32	0.89 ± 1.01	1.70 ± 1.97	0.014
Charlson index (points)	7.54 ± 2.68	7.07 ± 2.60	9.20 ± 2.31	0.001

*p*-Values are based on chi-square or Student’s *t*-tests. DBP, diastolic blood pressure; DM, diabetes mellitus; SBP, systolic blood pressure; URR, urea reduction rate.

**Table 2 nutrients-14-01659-t002:** Body composition and laboratory data of 90 hemodialysis patients.

Variable	Total (*n* = 90)	Survivors (*n* = 70)	Non-Survivors (*n* = 20)	*p*-Value
BMI (kg/m^2^)	25.18 ± 5.14	24.89 ± 4.76	26.19 ± 6.32	0.32
Exchange Na/K	1.36 ± 0.39	1.30 ± 0.35	1.57 ± 0.47	0.007
TBW (L)	35.88 ± 7.17	35.17 ± 6.72	38.38 ± 8.25	0.078
ECW (L)	19.13 ± 4.06	18.12 ± 3.26	22.70 ± 4.61	<0.001
ICW (L)	16.84 ± 4.48	17.14 ± 4.56	15.80 ± 4.15	0.24
ECM (kg)	28.26 ± 6.62	26.85 ± 5.74	33.19 ± 7.27	0.002
FFM (kg)	46.34 ± 9.87	45.65 ± 9.68	48.74 ± 10.41	0.22
BCM (kg)	18.07 ± 4.83	18.80 ± 4.96	15.55 ± 3.38	0.007
MM (kg)	23.42 ± 5.68	24.10 ± 5.85	21.10 ± 4.41	0.037
ECM/BCM ratio (points)	1.61 ± 0.41	1.46 ± 0.32	2.14 ± 0.23	<0.001
PA (°)	4.33 ± 0.97	4.56 ± 0.93	3.53 ± 0.66	<0.001
s-Albumin (g/dL)	3.76 ± 0.42	3.80 ± 0.43	3.50 ± 0.32	0.033
s-Transferrin (mg/dL)	169.40 ± 37.90	169.86 ± 35.29	167.90 ± 46.49	0.20
s-CRP (mg/dL)	0.95 ± 0.83	0.84 ± 0.81	1.32 ± 0.81	0.022
MIS (points)	8.46 ± 5.01	7.74 ± 4.99	10.95 ± 4.33	0.011
IL-6 (pg/mL)	4.19 ± 4.67	3.43 ± 4.38	6.84 ± 4.80	0.003

*p*-Values are based on Student’s *t*-tests. Values are expressed as mean ± SD. BCM, Body cell mass; BMI, body mass index; ECM, extracellular mass; ECM/BCM ratio, extracellular mass to body cell mass ratio; ECW, extracellular water; FFM, fat-free mass; ICW, Intracellular water; IL-6, interleukine-6; MIS, malnutrition-inflammation score; MM, muscle mass; PA, phase angle; s-CRP, serum C-reactive protein; TBW, total body water.

**Table 3 nutrients-14-01659-t003:** Univariate Cox proportional hazards analysis of all-cause mortality risk in hemodialysis patients.

Predictor Variable	Hazard Ratio (95%CI)	*p*-Value
Age (years)	1.046 (0.947–1.154)	0.377
Time on HD (months)	0.996 (0.985–1.008)	0.511
SBP (mmHg)	0.980 (0.958–1.003)	0.086
DBP (mmHg)	0.953 (0.920–0.987)	0.007
Vascular access (AVF/catheter)	0.223 (0.076–0.652)	0.006
URR (%)	1.046 (0.991–1.105)	0.105
Kt/V *urea* (*sp*)	1.443 (0.357–5.834)	0.607
BMI (kg/m^2^)	1.039 (0.967–1.117)	0.300
Exchange Na/K	2.859 (1.349–6.061)	0.006
TBW(L)	1.035 (0.981–1.092)	0.213
ECW (L)	1.220 (1.114–1.336)	<0.001
ICW(L)	0.917 (0.826–1.018)	0.105
ECM (kg)	1.090 (1.031–1.151)	0.002
FFM (kg)	1.013 (0.975–1.053)	0.500
BCM (kg)	0.836 (0.738–0.947)	0.005
MM (kg)	0.900 (0.820–0.987)	0.025
ECM/BCM ratio < 1.20 points	0.175 (0.051–0.606)	0.006
PA (°)	0.319 (0.199–0.512)	<0.001
s-Albumin (g/dL)	0.227 (0.085–0.610)	0.003
s-Transferrin (mg/dL)	0.990 (0.988 to 1.010)	0.790
MIS (points)	1.125 (1.036–1.221)	0.005
s-CRP (mg/dL)	1.607 (0.980–2.634)	0.06
IL-6 (median 3.1) (pg/mL)	0.206 (0.048 to 0.880)	0.033
Charlson index (points)	1.379 (1.161–1.638)	<0.001
Hospital admissions (number/per year)	1.405 (1.131–1.746)	0.002

AVF, arteriovenous fistula; BCM, body cell mass; BMI, body mass index; s-CRP, s-C-reactive protein; DBP, diastolic blood pressure; ECM, extracellular mass; ECM/BCM ratio, extracellular mass to body cell mass ratio; ECW, extracellular water; FFM, fat-free mass; ICW, intracellular water; IL-6, Interleukin-6; MIS, malnutrition-inflammation score; MM, muscle mass; PA, phase angle; SBP, systolic blood pressure; TBW, total body water; URR, urea reduction rate.

**Table 4 nutrients-14-01659-t004:** Multivariate Cox proportional hazards analysis of all-cause mortality risk in hemodialysis patients.

Predictor Variable	Hazard Ratio (95%CI)	*p*-Value
ECM/BCM ratio >1.20 points	5.078 (1.509 to 17.090)	0.009
IL-6 (median 3.1) (pg/mL) ^‡^	2.853 (1.076 to 7.565)	0.035
s-Albumin > 3.8 g/dL ⁌	0.225 (0.102 to 0.494)	<0.001
s-CRP (mg/dL)	1.107 (1.017 to 1.205)	0.019
Charlson index (points)	1.339 (1.130 to 1.587)	<0.001
Hospital admissions (number/per year)	0.858 (0.653 to 1.127)	0.272

^‡^ IL-6, interleukine-6. The median plasma IL-6 concentration (3.1 pg/mL) of the sample was used as the cut-off point for the Cox proportional hazards regression model. ⁌ s-Albumin concentration ≥ 3.8 g/dL was used according to the diagnostic criteria for PEW [4] in the Cox proportional hazard regression model. *p*-Value was adjusted for age, gender and time on hemodialysis (months). s-CRP, C-reactive protein; IL-6, interleukin-6; MIS, malnutrition-inflammation score.

## Data Availability

Not applicable.

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
