# Peer review of "The Extracellular Mass to Body Cell Mass Ratio as a Predictor of Mortality Risk in Hemodialysis Patients"

_nutrients, 2022, doi:10.3390/nu14081659_

Round 1
Reviewer 1 Report
The study decribes in a clear manner the importance of ECM/BCM ratio as a predicotr of mortality risk in hemodialysis.
I suggest minor corrections:
define "DM" in line 172.
correct "diabetes blood pressure" in "diastolic blood pressure" in the legend of Table 1.
I think s-albumin is correlated negatively and not positively (as mentioned in the text) with all-cause mortality (see line 215).
Author Response
Reviewer 1
The study describes in a clear manner the importance of ECM/BCM ratio as a predictor of mortality risk in hemodialysis.
I suggest minor corrections:
define "DM" in line 172.
correct "diabetes blood pressure" in "diastolic blood pressure" in the legend of Table 1.
I think s-albumin is correlated negatively and not positively (as mentioned in the text) with all-cause mortality (see line 215).
Answer. Thank you for the review and the comments suggested to improve our manuscript. The proposed changes are highlighted in yellow in the second version of the manuscript.
Reviewer 2 Report
The authors investigated the association between extracellular mass to body cell mass ratio and mortality risk in patients undergoing hemodialysis.
The topic of this study is interesting, but there are several concerns in this report.
First, I understand the authors investigated the extracellular mass to body cell mass ratio. However, this study was conducted as a prospective study. So, physicians should have adjusted patients’ dry weight if their extracellular mass to body cell mass ratio. Adjustment of dry weight is a fundamental measure against over extracellular volume in the hemodialysis field, and I do not think that participants maintained the same fluid valance throughout the observation period. If they did not adjust the dry weight of participants, it would cause ethical issues.
Second, participants who used catheters showed extremely poor prognosis in this study. It is well known that AVG is superior to the catheter in terms of life prognosis. The authors should show the reason why catheters were selected in the non-survivors. I assume that the catheter may have evoked inflammation and accelerated PEW in the non-survivors. The choice of blood access is a critical limitation in this study.
Third, the authors should not show the number of participants in the method part. Only inclusion and exclusion criteria should be shown here. In addition, line 96 “participants were undergoing at least 4 h dialysis by high flux dialysis” would be a kind of inclusion criteria. This statement should be shown in 2.1. Additionally, the authors should show the excluded patients due to advanced heart failure, and so on. In my opinion, patients who were prescribed NSAIDs and diuretics are not rare, so the authors should show the detail.
Fourth, categorial variables should be shown the number and percentage of each group.
For example, Gender Total 47 (52.2), Survivors 35 (50.0) non-survivors 12 (60.0) in Table 2. Moreover, there should be mistakes in the statistical analysis for catheter users.
Survivors 30 (42.9) non-survivors 18 (90.0). Therefore, the p-value should be <0.001.
Five, cardio-thoracic-ratio is very important to evaluate dry weight in patients undergoing hemodialysis rather than extracellular mass to body cell mass ratio. Nonetheless, the authors did not show the result.
Author Response
Reviewer 2
The authors investigated the association between extracellular mass to body cell mass ratio and mortality risk in patients undergoing hemodialysis.
The topic of this study is interesting, but there are several concerns in this report.
Answer. Thank you for the review and the valuable and constructive comments suggested to improve our manuscript. The proposed changes are highlighted in yellow in the second version of the manuscript.
First, I understand the authors investigated the extracellular mass to body cell mass ratio. However, this study was conducted as a prospective study. So, physicians should have adjusted patients’ dry weight if their extracellular mass to body cell mass ratio. Adjustment of dry weight is a fundamental measure against over extracellular volume in the hemodialysis field, and I do not think that participants maintained the same fluid valance throughout the observation period. If they did not adjust the dry weight of participants, it would cause ethical issues.
Answer. Thank you for your comments. During the study, dry weight adjustment in HD is a common and daily practice at the hemodialysis unit. The adjusted dry weight depends among other factors, on weight gained and interdialysis blood pressure control. Additionally, the presence of symptoms during the hemodialysis session such as hypotension, cramps or poor tolerance to ultrafiltration require post-dialysis dry weight adjustments. However, in our clinical practice, the dialysis dose is prescribed to reach the adjusted dry weight (symptom-free), although sometimes in patients with certain comorbidities (left ventricular hypertrophy with low ejection fraction or even patients with protein-energy wasting syndrome with marked hypoalbuminaemia (<3.5 g/dL) and inflammation), the prescribed adjusted target weight cannot be reached.
Therefore, as discussed above and from an ethical point of view, adjustment of dry weight, dialysis dose and monitoring of clinical manifestations both inter-dialysis, intra-dialysis and intra-dialysis are part of the standard care protocol in the hemodialysis unit.
Second, participants who used catheters showed extremely poor prognosis in this study. It is well known that AVG is superior to the catheter in terms of life prognosis. The authors should show the reason why catheters were selected in the non-survivors. I assume that the catheter may have evoked inflammation and accelerated PEW in the non-survivors. The choice of blood access is a critical limitation in this study.
Answer. Thanks for your appreciation. There is no doubt that vascular access, and in particular the arteriovenous fistula (AVF) compared to the central venous catheter has lower associated morbidity and mortality in HD patients. However, although the AVF is the first choice of vascular access in hemodialysis, in special circumstances such as thrombosis of the AVF that cannot be recovered until a new fistula is created, cardiovascular conditions that contraindicates the creation of an AVF, living donor renal transplant and/or the patient's express desire, the use of a central venous catheter is indicated, preferably a tunneled central venous catheter. Currently in Spain, data from the Spanish registry of Catalonia of almost 10,000 incident HD patients (1) showed that about 50% of HD patients started HD treatment through a central venous catheter. In prevalent HD patients as in our study, the causes of central venous catheter use are associated with the aforementioned special circumstances (stenosis or thrombosis of the fistula or lack of vascular access) according to the Spanish Society of Nephrology's guidelines for vascular access in hemodialysis (2). Therefore, given the high prevalence of tunneled catheters in clinical setting, it would not be ethically feasible to exclude these patients in the current study. Nevertheless, in accordance with your comments it has been included as a limitation in the second version of the manuscript.
1Roca-Tey R, Arcos E, Comas J, Cao H, Tort J; Catalan Renal Registry Committee. Vascular access for incident hemodialysis patients in Catalonia: analysis of data from the Catalan Renal Registry (2000-2011). J Vasc Access. 2015;16: 472-9.
2Ibeas J, Roca-Tey R, Vallespín J, et al. Spanish Clinical Guidelines on Vascular Access for Haemodialysis [published correction appears in Nefrologia (Engl Ed). 2019;39(6):680-682.
Third, the authors should not show the number of participants in the method part. Only inclusion and exclusion criteria should be shown here. In addition, line 96 “participants were undergoing at least 4 h dialysis by high flux dialysis” would be a kind of inclusion criteria. This statement should be shown in 2.1. Additionally, the authors should show the excluded patients due to advanced heart failure, and so on. In my opinion, patients who were prescribed NSAIDs.
Answer. Thank you for your suggestion. In the second version of the manuscript, in order to improve the understanding of the current study, a flow chart of the detailed recruitment and causes of exclusion has been included (see Figure 1).
Regarding the comment on line 96 (current line 115), it does not correspond to an inclusion criterion since all patients in the hemodialysis unit (included and excluded) were dialysed for at least 4 hours with high-flux semipermeable membranes.
As mentioned in the material and method section, advanced heart failure along with other pathologies and/or conditions (NSAIDs, corticosteroid or diuretics) that could interfere with nutritional, inflammatory, or fluid status were excluded from this study.
Fourth, categorial variables should be shown the number and percentage of each group.
For example, Gender Total 47 (52.2), Survivors 35 (50.0) non-survivors 12 (60.0) in Table 2. Moreover, there should be mistakes in the statistical analysis for catheter users.
Survivors 30 (42.9) non-survivors 18 (90.0). Therefore, the p-value should be <0.001.
Answer. Thank you for your valuable comments to improve our manuscript. Proposed changes can be found in the second version of the manuscript in table 1.
Five, cardio-thoracic-ratio is very important to evaluate dry weight in patients undergoing hemodialysis rather than extracellular mass to body cell mass ratio. Nonetheless, the authors did not show the result.
Answer. Thank you for your comment. Cardiothoracic ratio was not measured in this study. In addition, as indicated in the limitations of this study, cardiac imaging, functional tests and natriuretic peptides were not analysed. However, we appreciate your suggestion for improvement and plan to test and evaluate it as a marker of fluid overload in hemodialysis patients in future studies.
Reviewer 3 Report
INTRODUCTION
Please add the latest literature and clearly state what is known and what needs to be discovered
PATIENTS AND METHODS
Originally 190 people were included, what were the reasons why only 90 patients were included? Maybe it is worth attaching a diagram / drawing showing what the recruitment looked like.
The method does not cover all exclusion criteria from research. I mean the reasons why bioimpedance cannot be tolerated.
What were the exact causes of death? Please provide in numbers. Do the authors have such data? If so, they can be included in the complementary material.
What was the Kt / V formula used? (Daugridas? Cockroft-Gault?)
DISCUSSION
Add new references. Out of 40 references throughout the text, only 12 are from the last 5 years.
Table 1 and Table 3 show that patients with higher SBP and DBP had better survival. Is it true? If so, please refer to this result in the discussion.
Author Response
Reviewer 3
Thank you for your review and suggestions for improving our manuscript. The changes made are highlighted in yellow in the second version of the manuscript.
INTRODUCTION
Please add the latest literature and clearly state what is known and what needs to be discovered
Answer. Thank you for your comment. In accordance with your suggestions, the objective of the study has been modified in order to improve the comprehensibility of the manuscript. Furthermore, as shown in lines 71,72 of the introduction section, only a few studies have evaluated the ECM/BCM ratio in hemodialysis patients. Results from the Pubmed database using as keywords “ECM/BCM ratio AND dialysis”, show 4 records, of which the last published reference (Extracellular mass to body cell mass ratio as a potential index of wasting and fluid overload in haemodialysis patients. A case-control study. Clin Nutr. 2020;39(4):1117-1123. doi:10.1016/j.clnu.2019.04.021) belongs to the authors of the current paper. In addition, 10 of the 21 references included in the introduction section were published in the last 5 years. Also, the older references correspond to relevant studies or meta-analyses, as well as diagnostic consensus (e.g. reference number 9) or bioimpedance norms in hemodialysis (references 11,12), both of which are still valid criteria for application.
PATIENTS AND METHODS
Originally 190 people were included, what were the reasons why only 90 patients were included? Maybe it is worth attaching a diagram / drawing showing what the recruitment looked like.
Answer. Thanks for your suggestion. A flow chart of participant recruitment has been included in the second version of the manuscript (see Figure 1).
The method does not cover all exclusion criteria from research. I mean the reasons why bioimpedance cannot be tolerated.
Answer. Thanks. The second version of the manuscript includes possible causes for exclusion related to the performance of bioimpedance. In addition, some of these causes (e.g.leg amputation, pacemakers or joint replacements) are shown in Figure 1.
What were the exact causes of death? Please provide in numbers. Do the authors have such data? If so, they can be included in the complementary material.
Answer. Thank you for your comment. The causes of mortality are detailed in the results section in lines 187-189.
What was the Kt / V formula used? (Daugridas? Cockroft-Gault?).
Answer. Thank you for your comment. The second generation Kt/V urea (single pool) proposed by Daurgidas was used to measure dialysis adequacy. This concept has been included in the second version of the manuscript (Line 112).
DISCUSSION
Add new references. Out of 40 references throughout the text, only 12 are from the last 5 years.
Answer. As previously discussed in the introduction section, there are few studies to date that analyse the ECM/BCM ratio in hemodialysis (see comments above).
Table 1 and Table 3 show that patients with higher SBP and DBP had better survival. Is it true?. If so, please refer to this result in the discussion.
Answer. Thanks for your comment. As shown in table 1, the criteria for defining high blood pressure (≥140/90 mmHg) as a CV risk factor and all-cause mortality were not met by the SBP and DBP mean values found among the survivors and non-survivors of the study. Therefore, according to the data shown in Table 1, and results from the univariate analysis (Table 3), our results do not allow us to state that higher SBP and DBP levels are associated with better survival. In this study, it can be concluded that blood pressure control is a protective factor for mortality in hemodialysis, being one of the goals of dialysis (through ultrafiltration) to achieve dry weight.
Round 2
Reviewer 2 Report
The authors replied to my comments properly. I do not have comments anymore.